# The H2B deubiquitinase Usp22 promotes antibody class switch recombination by facilitating non-homologous end joining

Conglei Li[1], Thergiory Irrazabal[1], Clare C. So[1], Maribel Berru[1], Likun Du[2], Evelyn Lam[1], Alexandra K. Ling[1], Jennifer L. Gommerman[1], Qiang Pan-Hammarström [2] & Alberto Martin[1]

Class switch recombination (CSR) has a fundamental function during humoral immune response and involves the induction and subsequent repair of DNA breaks in the immunoglobulin (Ig) switch regions. Here we show the role of Usp22, the SAGA complex deubiquitinase that removes ubiquitin from H2B-K120, in the repair of programmed DNA breaks in vivo. Ablation of Usp22 in primary B cells results in defects in γH2AX and impairs the classical non-homologous end joining (c-NHEJ), affecting both V(D)J recombination and CSR. Surprisingly, Usp22 depletion causes defects in CSR to various Ig isotypes, but not IgA. We further demonstrate that IgG CSR primarily relies on c-NHEJ, whereas CSR to IgA is more reliant on the alternative end joining pathway, indicating that CSR to different isotypes involves distinct DNA repair pathways. Hence, Usp22 is the first deubiquitinase reported to regulate both V(D)J recombination and CSR in vivo by facilitating c-NHEJ.

[1] Department of Immunology, University of Toronto, Toronto, Ontario, Canada M5S 1A8. [2] Department of Laboratory Medicine, Karolinska Institutet, Stockholm SE14186, Sweden. Correspondence and requests for materials should be addressed to A.M. (email: alberto.martin@utoronto.ca)

Antibody diversification is essential for vertebrates to prevent and eradicate infections[1–3]. Primary diversification, i.e., V(D)J recombination, creates the primary antibody repertoire and forms the antigen binding domain of the antibody[1]. This process involves the generation and subsequent repair of RAG1/2-induced double-stranded DNA breaks (DSBs) at specific recombination signal sequences that flank each V, D, and J coding segment within the variable region[1]. Successful V(D)J rearrangement is essential for expression of the B-cell receptor (BCR) and progression through B-cell development[1]. Class switch recombination (CSR) is a secondary diversification process that drives the generation of antibodies of various isotypes (e.g., from IgM to IgG or IgA)[2,3]. The isotype of an antibody controls its effector functions. CSR is initiated by activation-induced deaminase (AID), which catalyzes the deamination of deoxycytidines (i.e., $dC{\to}dU$) in the switch regions that are upstream of each constant region within the immunoglobulin heavy chain locus, leading to a dU:dG mismatch[3,4]. This mismatch is further processed by the mismatch repair and base excision repair pathways, resulting in the production of staggered DSBs[5].

DSBs initiated by either RAG1/2 or AID induce a cascade of DNA damage signaling, in which phosphorylation of H2AX on serine 139 ($\gamma$H2AX) has a critical function by recruiting various DNA repair factors (e.g., 53BP1 and RIF1) to DSB regions[6–8]. During V(D)J recombination, repair of DSBs is mediated by the classical non-homologous end joining (c-NHEJ) pathway to produce a functional V-region exon[1]. During CSR, DSBs within two switch regions are repaired primarily by the c-NHEJ machinery and to a lesser degree by the alternative end joining (A-EJ) pathway[9–11], leading to a replacement of the μ constant region with another constant region downstream of the recombined V(D)J segment. c-NHEJ involves ligation of DNA ends with little to no homology, whereas A-EJ, which is not well defined, involves regions of microhomology (MH) for end ligation[11,12].

Studies have demonstrated that ubiquitination is a critical mechanism in the regulation of signaling transduction in many biological processes, including immune responses[13], but the functions of deubiquitinases in B cells are not clear. In our previous efforts of searching for new factors that are involved in the CSR process, components of the SAGA (Spt-Ada-Gcn5-acetyltransferase) deubiquitinase complex were identified in a whole-genome RNA interference screen as being required for CSR in the CH12F3-2 (CH12) B-cell line[14]. The SAGA complex deubiquitinase Usp22 removes ubiquitin from histone H2B at lysine 120, while addition of this histone mark is catalyzed by the E3 ubiquitin ligase RNF20/RNF40 heterodimer[15,16]. H2B mono-ubiquitination at lysine 120 (hereafter referred to as H2Bub) has been implicated in the DNA damage response, as the addition of a ubiquitin moiety to H2B was proposed to induce chromatin relaxation, thereby increasing the accessibility of DNA repair factors to DNA damage[16,17]. Our previous work with CH12 cells demonstrates that Usp22 is required for CSR in vitro[14]. However, as CH12 cells are a lymphoma-derived cell line[18], investigation of Usp22 function in more physiological conditions, such as B-cell development and CSR in vivo, is required.

To assess whether Usp22 is involved in the repair of programmed DSBs in B cells in vivo, we here generate Usp22[flox/flox] mice and cross these mice with CD19-cre or Mb1-cre to knock out Usp22 specifically in B cells. B-cell-specific Usp22 KO mice have defects in CSR to various Ig isotypes, but not to IgA. IgG CSR is primarily mediated by the c-NHEJ pathway, whereas IgA CSR is more dependent on alternative end joining, indicating that CSR to different isotypes may involve distinct DNA repair pathways. Early pro-B ablation of Usp22 results in the blockade of B-cell development and defects in V to DJ recombination,

supporting other findings that Usp22 promotes c-NHEJ. In summary, by creating the first KO mice of the SAGA complex, we show that Usp22 has an important function in repairing DSBs that occur during B-cell development.

## Results

**Generation of B-cell-specific Usp22 KO mice.** Previous work with the CH12 cell line suggested that Usp22 has a critical role in DSB repair pathways that govern CSR (Fig. 1a). To test whether Usp22 is necessary to repair programmed DSBs that occur during B-cell development in vivo (i.e., V(D)J recombination and CSR), we employed the *Cre-LoxP* system to generate Usp22 conditional KO mice, as Usp22 is essential for mouse embryogenesis[19]. Usp22[flox/flox] mice (Fig. 1b) were crossed with CD19-cre mice[20] to knock out Usp22 in pre-B cells. Quantitative PCR (qPCR) analysis showed that Usp22 messenger RNA was reduced in spleen B cells from CD19-cre-Usp22 KO mice compared with wild-type (WT) littermates (Fig. 1c). Lipopolysaccharide (LPS) stimulation resulted in a decrease in Usp22 mRNA level in WT B cells when compared to the unstimulated cells (Supplementary Fig. 1a). Consistent with its role in deubiquitinating H2B, we found that the level of H2Bub was markedly increased in splenic B cells from CD19-cre-Usp22 KO mice (Fig. 1d and Supplementary Fig. 1b). We found that deletion of Usp22 did not markedly affect various spleen B-cell subsets (Fig. 1e and Supplementary Fig. 1c-e).

To evaluate whether B-cell deletion of Usp22 affects CSR in vivo, we first assayed the levels of various Ig isotypes in the serum of Usp22 KO or control mice. We found that the levels of polyclonal IgG1 and IgG3 isotypes were reduced and IgM was increased in the serum of naive CD19-cre-Usp22 KO mice compared with naive WT littermates (Fig. 1f). Interestingly, no defect in the serum IgA level was observed in CD19-cre-Usp22 KO mice (Fig. 1f). As IgA has an essential role in the control of intestinal microbiota[21], we also tested the level of polyclonal IgA in the feces and also found no defect in fecal IgA levels in CD19-cre-Usp22 KO mice (Fig. 1f). These data demonstrated that Usp22 is critical for H2B deubiquitination in primary B cells and is required for CSR to IgG subclasses but not to IgA, under homeostatic conditions.

**Usp22 KO mice exhibited defects in IgG but not IgA responses.** To further evaluate the role of Usp22 on CSR, we first utilized Nitrophenyl Chicken Gamma Globulin (NP-CGG) as a model antigen to immunize mice[22]. At day 22 post-NP immunization, we found that anti-NP IgG1 levels were reduced, whereas anti-NP IgM levels were increased in CD19-cre Usp22 KO mice, compared with WT littermates (Supplementary Fig. 2a). We observed that the number of NP-specific IgG[+] antibody secreting cells was reduced in the spleen of Usp22 KO mice, compared with WT littermates (Fig. 2a). To measure antigen-specific IgA responses, mice were infected with rotavirus[23], as IgA is the predominant Ig isotype induced by this pathogen in the small intestine and contributes to fecal rotavirus antigen clearance[24]. Enzyme-linked immunosorbent assay (ELISA) analysis of fecal anti-rotavirus IgA revealed no defect in the generation of anti-rotavirus IgA in the CD19-cre-Usp22 KO mice compared with WT littermates (Fig. 2b). Furthermore, we found no defect in IgA[+] rotavirus-specific antibody secreting cells in the lamina propria or bone marrow (BM), or in fecal rotavirus antigen clearance in CD19-cre-Usp22 KO mice compared with WT littermates (Fig. 2c and Supplementary Fig. 2b). Hence, expression of Usp22 by B cells is required to mount antigen-specific IgG responses, but not antigen-specific IgA responses.

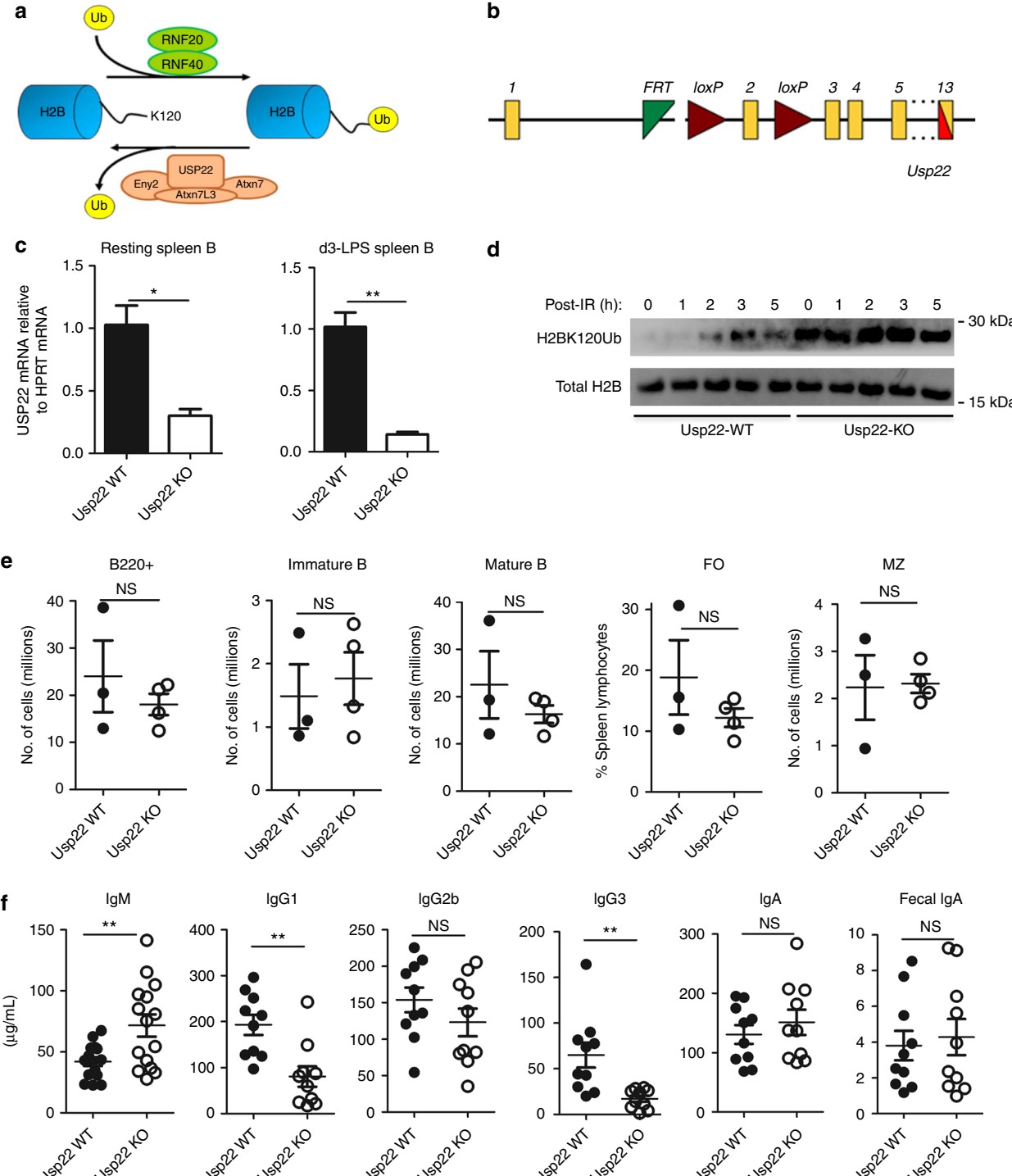

**Fig. 1** Spleen B-cell profiles and total Ig isotype analysis of Usp22 KO or WT mice. **a** Schematic showing the dynamic regulation of histone H2B ubiquitination (Ub). Addition of ubiquitination to H2B is mediated by RNF20/RNF40, while the removal of ubiquitination from H2B is mediated by SAGA complex that is composed of Usp22, Eny2, Atxn7, and Atxn7L3. **b** The structure of targeted *usp22* locus. The protein-encoding exon 2 (133 bp) of the *usp22* gene was flanked by *loxP* sites (triangles). Usp22^{flox/flox} mice are bred to CD19-cre or Mb1-cre mice to delete exon 2, leading to disruption of *usp22* gene expression specifically in B cells. **c** Spleen B cells were purified from CD19-cre-Usp22 KO mice or WT littermates. qPCR analysis of Usp22 mRNA was performed with resting spleen B cells or B cells that were stimulated with LPS for 3 days ex vivo ($n = 3$ mice per group). Data represent two independent experiments. **d** Spleen B cells from CD19-cre-Usp22 KO and WT littermate mice were stimulated with LPS for 2.5 days, exposed to 8 Grays of γ-radiation, and then collected at various time points for western blot analysis of H2B monoubiquitination (H2BK120Ub). Data represent four independent experiments. **e** Absolute numbers analysis of spleen B cells from CD19-cre-Usp22 KO and WT littermates. FO, follicular B cells; MZ, marginal zone B cells. Data represent two independent experiments each with three to four mice per group. **f** Serum analysis of various Ig isotypes in unimmunized CD19-cre-Usp22 KO and WT littermates, as well as fecal IgA from the same mice ($n = 10$ mice per group). Data in **c**, **e**, and **f** were presented as mean ± SEM and were analyzed using two-tailed unpaired Student's *t*-test. *$p < 0.05$ and **$p < 0.01$; NS, not significant

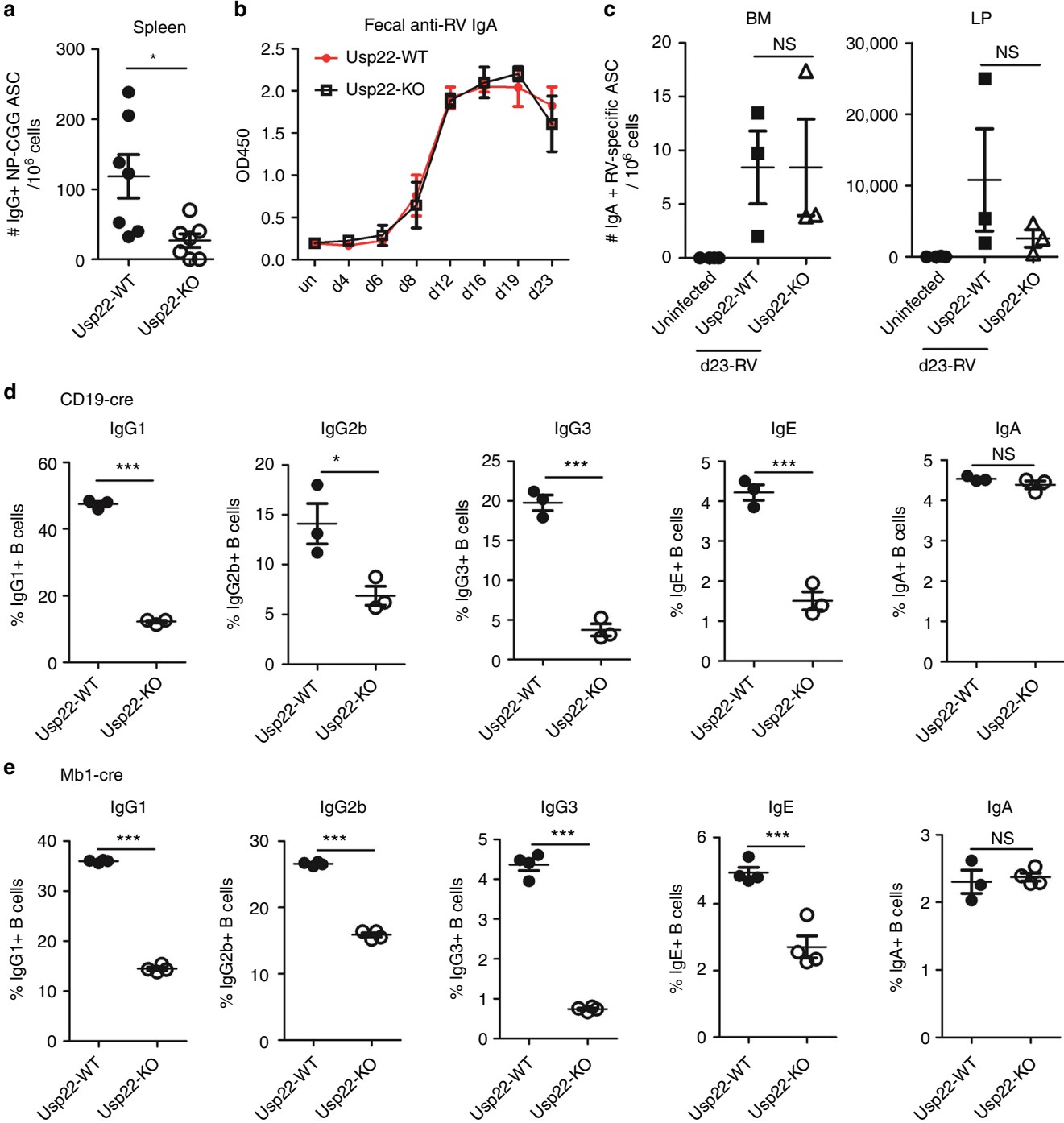

**Fig. 2** Usp22 KO mice exhibited defective IgG/IgE CSR but not IgA CSR. **a** NP$_{25}$-CGG plus alum was intraperitoneally injected into CD19-cre Usp22 KO or WT littermate mice. At d22 post immunization, mice were subjected to NP-specific IgG$^+$ ELISPOT assay to measure the amount of NP-specific IgG$^+$ antibody secreting cells (ASC) in the spleen ($n = 7$ mice per group). Data were combined from two independent experiments. **b** Rotavirus (RV) was orally administered to CD19-cre-Usp22 KO or WT littermates. Fecal samples were collected at the indicated time-points post-RV infection, followed by fecal supernatant preparation. The 1:2 dilution of fecal supernatant was used for fecal anti-RV IgA ELISA assay. Data was analyzed by two-way ANOVA. **c** ELISPOT assays were performed at d23 post-RV infection, to measure the numbers of IgA$^+$ RV-specific antibody secreting cells in lamina propria (LP) and bone marrow (BM), respectively. Data in **b** and **c** represent two independent experiments each with three mice per group. **d** Splenic B cells from CD19-cre-Usp22 KO or WT littermates were ex vivo induced to switch to IgG1, IgG2b, IgG3, IgE, and IgA ($n = 3$ mice per group). **e** Same as **d**, except that Mb1-cre Usp22 KO mice were used ($n = 3$–4 mice per group). Data in **d** and **e** represent three independent experiments; data in **a** and **c**–**e** were analyzed using two-tailed unpaired Student's $t$-test. Data were presented as mean ± SEM. *$p < 0.05$ and ***$p < 0.001$; NS, not significant

To determine whether the defect in CSR in Usp22-deficient B cells is cell intrinsic, we induced CSR to various Ig isotypes ex vivo using purified spleen B cells. We found that CSR to IgG1, IgG2b, IgG3, and IgE were markedly reduced in spleen B cells from CD19-cre-Usp22 KO mice, compared with littermate controls (Fig. 2d and Supplementary Fig. 2c). Again, surprisingly, IgA CSR was normal in CD19-cre-Usp22 KO splenic B cells (Fig. 2d). To eliminate the possibility that this phenomenon was due to the CD19-cre transgene, we crossed Usp22$^{flox/flox}$ mice with Mb1-cre mice to delete Usp22 in pro-B cells and induced CSR ex vivo[25]. We again observed defects in IgG and IgE CSR, but not in IgA CSR in splenic B cells from Mb1-cre-Usp22 KO mice (Fig. 2e). Collectively, these data suggest that Usp22 is required for switching to IgG and IgE, but not to IgA, in splenic B cells. In addition, we isolated B cells from lymph nodes (LNs) and found that CD19-cre-Usp22 KO LN B cells were also defective in IgG1 CSR ex vivo when compared with littermate control (Supplementary Fig. 2d). Interestingly, we observed that Usp22 KO LN B cells exhibited a minor defect in IgA CSR (Supplementary Fig. 2d). The minor difference in IgA CSR between spleen and LN B cells may be due to the possibility that mature B cells are more predominant in LN than in the spleen.

As Mb1-cre-mediated gene deletion can occur in early pro-B cells before V(D)J rearrangement[25], we tested whether Usp22 is involved in the repair of DSBs that occur during V(D)J recombination. Hence, we analyzed B-cell profiles in the BM using the "Hardy" staining scheme[26]. The early B-lineage cells (Hardy Fractions A and B) actively undergo D to J rearrangement (in Fraction B); up to one quarter of these early B cells also undergo V to (D)J recombination in their IgH alleles[27,28]. We observed that early pro-B cells (B220$^+$CD43$^+$), especially the Fraction B subset (Fr B: CD43$^+$/B220$^+$/CD24$^+$/BP-1$^-$), were markedly increased in the BM of Mb1-cre-Usp22 KO mice, compared with WT littermates (Supplementary Fig. 3a-c), indicative of a blockade in B-cell development in Usp22 KO mice. We evaluated V(D)J recombination at the molecular level and found that distal $V_H$ to $DJ_H$ segment recombination, but not proximal $V_H$-to-$DJ_H$ or D-to-J recombination, seemed to be impaired in pro-B cells from Mb1-cre-Usp22 KO mice (Supplementary Fig. 3d-e). These data suggest that Usp22 is involved in the long-range V(D)J recombination and B-cell development.

**Impaired γH2AX levels in Usp22 KO spleen B cells.** To investigate the potential mechanisms for IgG/IgE CSR defects in Usp22-deficient B cells, we first assessed whether AID expression was impaired, as AID is essential for CSR[4]. We found that AID was increased at both mRNA and protein levels in spleen B cells from CD19-cre-Usp22 KO mice compared with WT littermates (Fig. 3a and Supplementary Fig. 4a), which might be due to the role of Usp22 in regulating the gene transcription[29]. However, these data are inconsistent with a reduction in CSR to IgG/IgE in Usp22-deficient B cells. One possibility that could explain the CSR defect to IgG/IgE, but not IgA in Usp22-deficient B cells, is the reduction of germline transcripts in γ,ε switch regions, but not α switch region. By evaluating the germline transcripts at day 2 post-CSR stimulation, we found comparable levels of μ, γ1, and α germline transcripts between Usp22 KO and WT splenic B cells (Supplementary Fig. 4b-c). To test whether AID function at the μ and γ1 switch region was affected, we sequenced the regions upstream of the μ and γ1 switch region, respectively. However, we found no significant difference in AID mutation frequency in splenic B cells between CD19-cre-Usp22 KO and WT group (Table 1), which is consistent with our previous studies with Usp22 KO CH12 cells[14]. Together, these data indicate that differential effects of Usp22 deficiency on IgG/IgE vs IgA CSR in

primary B cells are independent of AID activity, suggesting that repair of DSBs downstream of AID is different when leading to IgG/IgE CSR than IgA.

γH2AX has a critical role in DSB repair by recruiting various DNA repair factors to DSB regions[6–8], and it has been shown that premature degradation of γH2AX signaling leads to impaired DNA damage repair[30]. To test whether enhanced H2Bub has an impact on γH2AX kinetics in mice, splenic B cells were stimulated with LPS and then exposed to γ-radiation, followed by γH2AX immunofluorescent staining. We found that γH2AX kinetics were altered in CD19-cre-Usp22 KO B cells, with a marked decrease in γH2AX four hours post irradiation (Fig. 3b). We also observed a premature degradation of γH2AX in Usp22 KO group under conditions that lead to IgA CSR (Fig. 3b), which is consistent with our previous data in IgA-switched CH12 cells[14]. As a defect in γH2AX signaling could manifest in a defect in DSB repair that can impair cell cycle progression during CSR, we evaluated whether the cell cycle was altered in Usp22 KO primary B cells. By pulsing cells with Edu and then staining with propidium iodide (PI), we found that LPS-stimulated spleen B cells from CD19-cre-Usp22 KO mice exhibited an increased proportion of S-phase cells and a decreased proportion of G1 and G2-M phases compared with controls (Fig. 3c and Supplementary Fig. 4d). Consistent with this finding, proliferation assays showed that LPS-stimulated Usp22-KO B cells proliferated more quickly than WT B cells (Fig. 3d). However, the IgG/IgE CSR defects in Usp22-deficient spleen B cells are unlikely due to alterations in cell cycle, as CSR to IgG1 is defective in CD19-cre-Usp22 KO B cells at various stages of division (Fig. 3e and Supplementary Fig. 4e). Collectively, these data show that Usp22 deficiency results in premature degradation of γH2AX signaling and a potentially disrupted G$_1$–S checkpoint, which may manifest in impaired DNA repair during CSR.

**IgA CSR is more dependent on A-EJ than IgG CSR.** As V(D)J recombination requires c-NHEJ[31], the impaired pro-B-cell development in Mb1-cre-Usp22 KO mice suggests that Usp22 is critical for c-NHEJ. The repair of DSBs during CSR is mediated by both the c-NHEJ pathway and A-EJ pathway[9,10]. Unlike DSBs repaired by A-EJ, DSBs repaired by c-NHEJ typically have little to no MH between the donor and acceptor switch regions[11]. As Usp22 KO splenic B cells exhibited defective CSR to IgG and IgE but not IgA, we hypothesized that CSR to various Ig isotypes may rely on different DNA repair pathways to various degrees. To test this hypothesis, we examined switch junctions from IgG1- vs IgA-switched spleen B cells, as analysis of MH usage at switch junctions can reflect the DNA repair pathways that were utilized during CSR. Sequencing of IgG1-switched spleen B cells revealed that the MH length between the μ and γ1 switch regions is comparable between CD19-cre-Usp22 KO and WT littermates (MH: 2.4 vs 2.2, $P = 0.75$) (Fig. 4a), although a trend of increased 6 + bp MH usage was observed in Usp22-KO IgG1 B cells (9%) when compared with Usp22-WT B cells (2%). In contrast, we found a significantly longer MH between the μ and α switch regions in IgA-switched CD19-cre-Usp22 KO spleen B cells, compared with WT littermates (MH: 3.0 vs 1.9, $P < 0.05$) (Fig. 4b). The percentage of insertions was slightly decreased in Usp22 KO B cells when compared with WT control (Fig. 4a,b). The pattern of breakpoint locations, which were distributed across the switch regions, were comparable between Usp22-WT and -KO groups (Fig. 4c,d).

The increased MH usage in μ−α switch regions but not μ−γ1 switch regions in Usp22-deficient B cells has been observed before in DNA Ligase IV- and DNA-PKcs-deficient human B cells (i.e., defective in c-NHEJ)[32,33], and suggests that CSR to IgA

employs A-EJ more efficiently when c-NHEJ is impaired, while CSR to IgG1 does not. These results further suggest that Usp22 promotes the c-NHEJ but not the A-EJ pathway. To further validate the distinct dependence of IgG and IgA CSR on c-NHEJ vs A-EJ, we evaluated the surface Ig isotype in spleen B cells from 53BP1-deficient mice, which are deficient in c-NHEJ[34], after ex vivo CSR stimulation. We found that *53BP1*$^{-/-}$ B cells exhibited a less severe defect in IgA CSR (~ 60% reduction, compared with controls) than CSR to other Ig isotypes (~ 80–90% reduction) (Fig. 5a). In addition, we used the SUV4-20 inhibitor

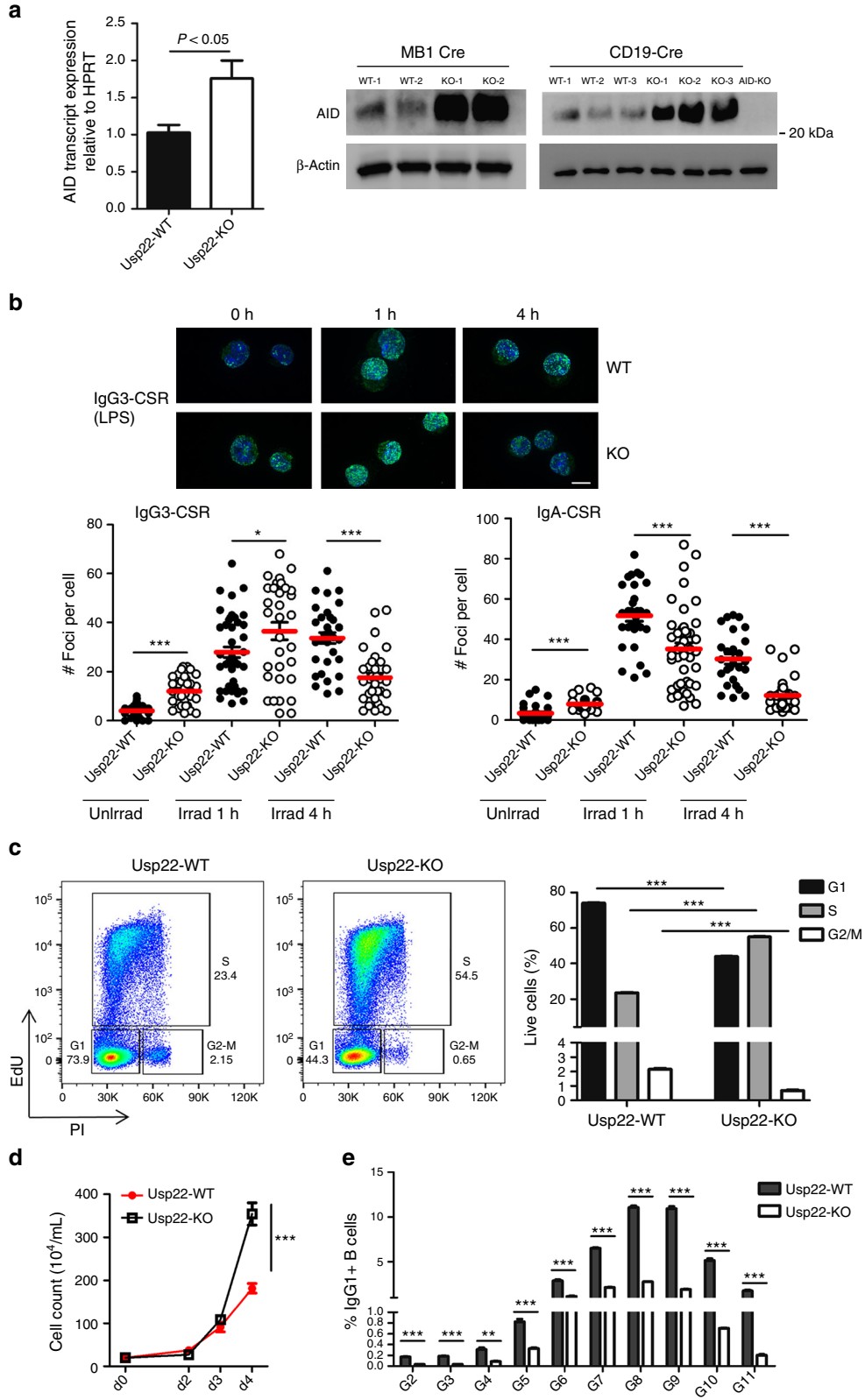

A-196 (A-197 as the control compound) to suppress c-NHEJ during ex vivo CSR of WT spleen B cells[35]. Inhibition of c-NHEJ caused a defect in IgG1 CSR (~ 50% reduction), but not IgA CSR (Fig. 5b).

To determine why CSR to IgA is more dependent on the A-EJ pathway than switching to other isotypes, we analyzed the sequence homology between switch regions within the mouse IgH locus. We reasoned that since A-EJ utilizes microhomologies during the reaction, recombining sequences with increased homology might favor A-EJ over c-NHEJ. We found that in both C57BL/6 and BALB/c mice, a higher degree of homology was observed between Sμ and Sα, when compared with that between Sμ and Sγ1, Sγ2b, and Sγ3 switch regions (Fig. 5c,d and Supplementary Fig. 5-6). It is notable that Sμ and Sε switch regions also had a relatively high degree of nucleotide identity, although to a lesser degree as compared with the Sμ and Sα switch regions. Hence, the high degree of primary DNA sequence identity between Sμ and Sα switch regions might explain why A-EJ is more efficient during IgA CSR at the molecular level.

## Discussion

By generating the first KO mouse of the SAGA complex, we identified Usp22 as a new factor that is essential to repair programmed DNA breaks that occur during B-cell ontogeny. We demonstrate that Usp22 promotes c-NHEJ and CSR in vivo. Interestingly, B-cell-specific ablation of Usp22 resulted in defective CSR to various IgG/IgE isotypes but not to IgA. We further demonstrated that CSR to IgG/IgE is primarily mediated by c-NHEJ, whereas CSR to IgA is more dependent on MH-mediated A-EJ. Indeed, consistent with a previous report that measured serum Ig isotypes in $53BP1^{-/-}$ mice[36], we found by measuring surface Ig isotypes that $53BP1^{-/-}$ B cells exhibited a less severe defect in IgA CSR than CSR to other Ig isotypes. In addition, we observed that chemical inhibition of c-NHEJ has strong effects on IgG CSR but not IgA CSR, which is similar to the CSR phenotype in Usp22-deficient B cells. The difference in IgA CSR between Usp22 KO mice and $53BP1^{-/-}$ mice may be due to the fact that c-NHEJ is essentially null in $53BP1^{-/-}$ mice but is likely only reduced in Usp22 KO mice. In addition to IgA CSR, it has been reported that A-EJ pathways are also preferentially utilized for CSR to IgD[37,38]. As μ and α switch regions from both human and mice share higher sequence identity compared with other switch regions, this might be the reason for the increased dependence of A-EJ for IgA CSR, which in turn would suggest that primary DNA sequences can partly dictate the usage of DNA repair pathways.

Ubiquitination-mediated signaling transduction has important roles in many biological processes, including various aspects of immune functions[13]. In contrast to ubiquitylating enzymes, the roles of deubiquitinases are less well studied, especially in the

context of B cells. Histone H2B has been implicated in DNA damage response, as H2Bub at lysine 120 is thought to induce changes in chromatin conformation, making damaged DNA accessible by various repair factors[16,17]. Usp22 has been shown to deubiquitinate both histone H2A and H2B in biochemical assays[39,40]. A recent report showed that H2Bub levels are not increased in Usp22-deficient HEK293T cells due to redundancy with other deubiquitinases[41]. However, we found that Usp22 deletion in primary spleen B cells resulted in a marked increase in H2Bub, which is consistent with our findings in Usp22-KO CH12 cells[14], indicating that Usp22 is essential for H2B deubiquitination in B cells, and that other deubiquitinahave significant backup roles to Usp22 in these cells.

V(D)J recombination is essential for the surface expression of BCR and T-cell receptor[1]. We found that Usp22 deletion in early B cells mediated by Mb1-cre caused a block at the Fraction B stage during B-lymphopoesis. At the molecular level, we observed that long-range V-(D)J rearrangement seems to be impaired in the Mb1-cre-Usp22 KO mice, which is similar to the defective V-(D)J recombination phenotype observed in Pax5-, IL-7R-, or 53BP1-deficient mice[27,42–44], or patients with NIPBL mutations[45]. Nevertheless, more work needs to be carried out to establish a role for Usp22 in V(D)J recombination, and whether the defect in B cell development is due to altered cell cycle and/or cell death in the Usp22 KO mice. It also remains to be investigated why distal vs proximal V-to-DJ recombination has different requirements for certain c-NHEJ factors, such as 53BP1 and Usp22.

Phosphorylation of H2AX has a critical role in the DNA damage response by recruiting various DNA repair factors to damage regions[6–8], and it has been shown that premature degradation of γH2AX signaling impairs DNA repair[30]. We found that ablation of Usp22 in B cells resulted in a defect in irradiation-induced γH2AX levels at later time-points (i.e., 4 h post irradiation)[14]. The potential mechanisms by which excessive H2Bub affects γH2AX signaling are unknown. It is possible that excessive H2Bub inhibits the recruitment/stability of γH2AX kinase (i.e., ATM and DNAPK) via steric hindrance and/or preferentially recruits certain γH2AX phosphatases to sites of DNA damage.

IgA is the most abundant antibody isotype in the body and has an essential role in the control of pathogen invasion and commensal bacteria at mucosal surfaces, such as the gastrointestinal tract[46]. Many recent studies including our own have demonstrated that dysregulation of the microbiome are linked to many diseases, such as inflammatory bowel disease, neurologic diseases, and various cancers[47,48]. In our previous report, we found that knockdown or KO of Usp22 caused a defect in IgA CSR in the CH12 cell lines[14]. Here, by generating B-cell-specific Usp22 KO mice, we found that Usp22 ablation led to defects in CSR to

**Fig. 3** Impaired γH2AX levels in spleen B cells from CD19-cre-Usp22 KO mice. **a** AID mRNA measured by qPCR of d3-LPS stimulated spleen B cells from CD19-cre-Usp22 KO or WT littermates (left panel; n = 5 mice per group). AID blot analysis of d3-LPS stimulated spleen B cells from Mb1-cre or CD19-cre Usp22 KO or their WT littermates (right panel; n = 2–3 mice per group); AID-KO cells were used as control and data represent two or three independent experiments. **b** Spleen B cells from CD19-cre-Usp22 KO and WT littermates were stimulated for IgG3 CSR (by LPS) or for IgA CSR (by LPS+ IL-4+ TGF-β+ IL-5+ anti-IgD dextran) for 2.5 days, exposed to γ-radiation, and then collected for γH2AX immunofluorescent staining. Line within micrograph represents 10 μm. The number of γH2AX foci per cell was analyzed 1 or 4 h post irradiation for Usp22-WT (black circles) and CD19-cre-Usp22-KO (white circles) (bottom panel), and representative images for IgG3-CSR group (γH2AX in green and DAPI in blue) are shown in the top panel. Approximately 30–50 cells were analyzed per sample and data represent two or three independent experiments. **c** Representative flow cytometry plots (left panel) and compiled cell cycle analysis (right panel) of Usp22-WT and CD19-cre-Usp22-KO spleen B cells. Cells were stimulated with LPS for 2.5 days and then incubated with EdU for 4 h, before FACS staining (n = 3 samples per group). **d** Purified splenic B cells from CD19-cre Usp22-KO or WT littermates was stimulated with LPS and growth of these cells was monitored over time (n = 3 samples per group). Data were tested by two-way ANOVA. **e** CFSE-pulsed spleen B cells from CD19-cre-Usp22 KO or WT littermates were stimulated by LPS plus IL-4 for CSR to IgG1 for 4 days. The gating (G2 to G11) is based on the intensity of CFSE and the percentage of IgG1 switch per gate is summarized (n = 3 samples per group). Data in **c–e** represent two independent experiments; data in **a–c** and **e** were analyzed using two-tailed unpaired Student's t-test. Data were presented as mean ± SEM. *p < 0.05, **p < 0.01, and ***p < 0.001

**Table 1 Frequency of mutations upstream of switch μ or γ1 regions**

| | Usp22 WT | | Usp22 KO | |
|---|---|---|---|---|
| | S'μ[c] | S'γ1[d] | S'μ[c] | S'γ1[d] |
| Sequences (#) | 54 | 47 | 50 | 35 |
| Nucleotides sequenced (#) | 30,742 | 24,974 | 26,404 | 17,562 |
| Mutations (#) | 14 | 5 | 16 | 3 |
| Mutation frequency[a] | $4.55 \times 10^{-4}$ | $2.00 \times 10^{-4}$ | $6.05 \times 10^{-4}$ | $1.71 \times 10^{-4}$ |
| Mutations at G/C (%) | 86 | 100 | 100 | 100 |
| AID hotspot mutations (%)[b] | 79 | 100 | 88 | 100 |
| Deletions/insertions (#) | 0 | 0 | 0 | 1 |

[a]Frequency is defined as unique mutations/nucleotide sequenced. [b]Mutations at W$\underline{RC}$ and $\underline{GY}$W motifs, where W = A/T, R = A/G, and Y = T/C. [c]Cells stimulated to IgA ($n = 4$ mice per group). [d]Cells stimulated to IgG1 ($n = 3$ mice per group).

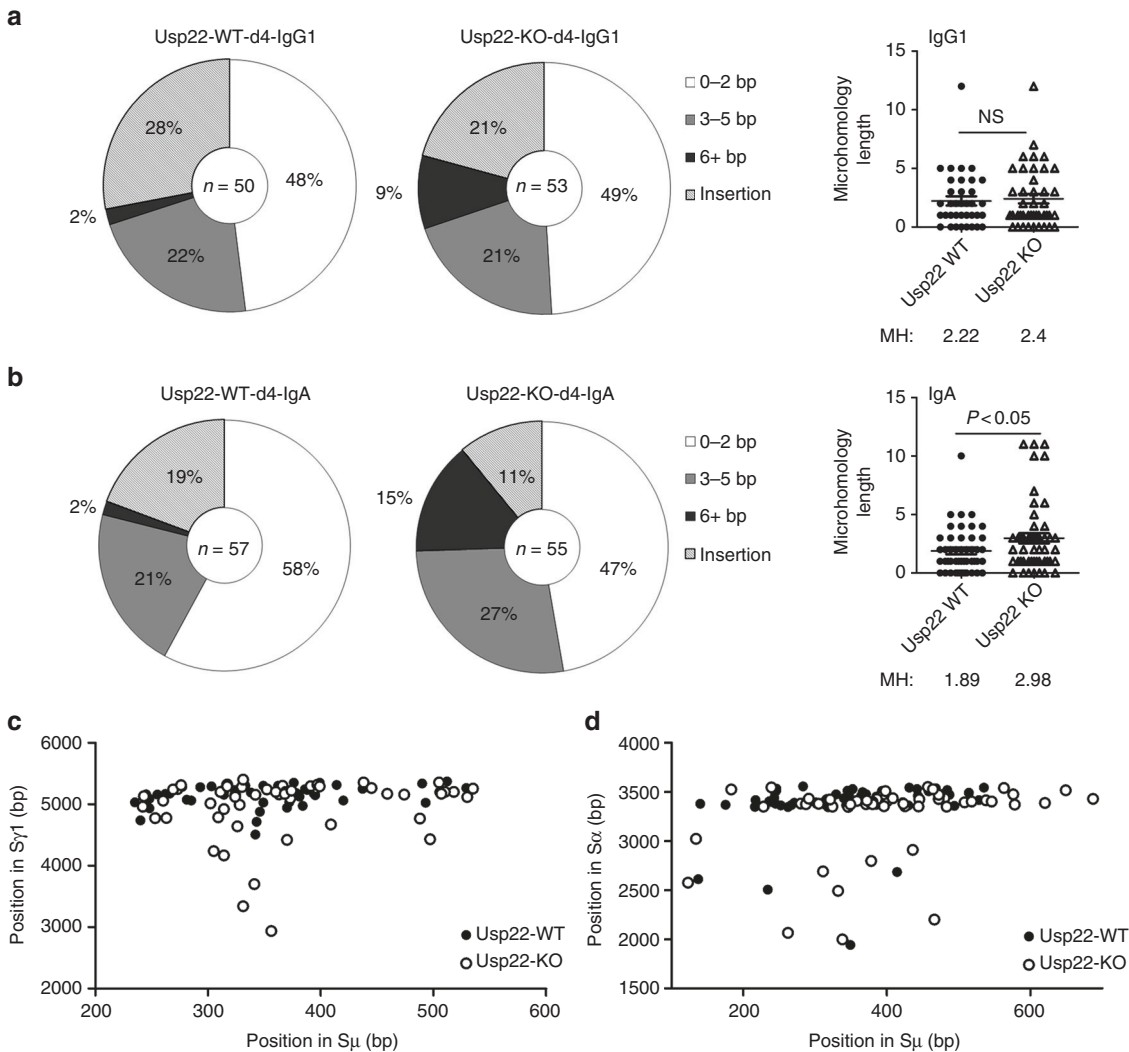

**Fig. 4** Microhomology analysis of switch junctions of IgG1- or IgA-switched spleen B cells. Spleen B cells isolated from CD19-cre-Usp22 KO or WT littermates were induced for IgG1 (**a** and **c**) or IgA (**b** and **d**) CSR ex vivo for 4 days. IgG1 and IgA switch junctions were cloned from these cells, and subjected to sequencing. Approximately 50–60 unique switch junction sequences were analyzed per group. **a** Sμ/Sγ1 or **b** Sμ/Sα junctions were categorized based on their microhomology usage: near-blunt joins (0–2 bp), joins with microhomology (3–5 bp or 6 + bp), or insertions. The microhomology length (MH) used by each unique switch junction sequence are summarized (right panel). **c** Distribution of breakpoints across μ or γ1 switch regions for IgG1-switched Usp22-WT or -KO cells. **d** Distribution of breakpoints across μ or α switch regions for IgA-switched Usp22-WT or –KO cells. Data in **a**–**d** were combined from three mice per group; data in **a** and **b** were presented as mean ± SEM and were analyzed using two-tailed unpaired Student's *t*-test. NS, not significant

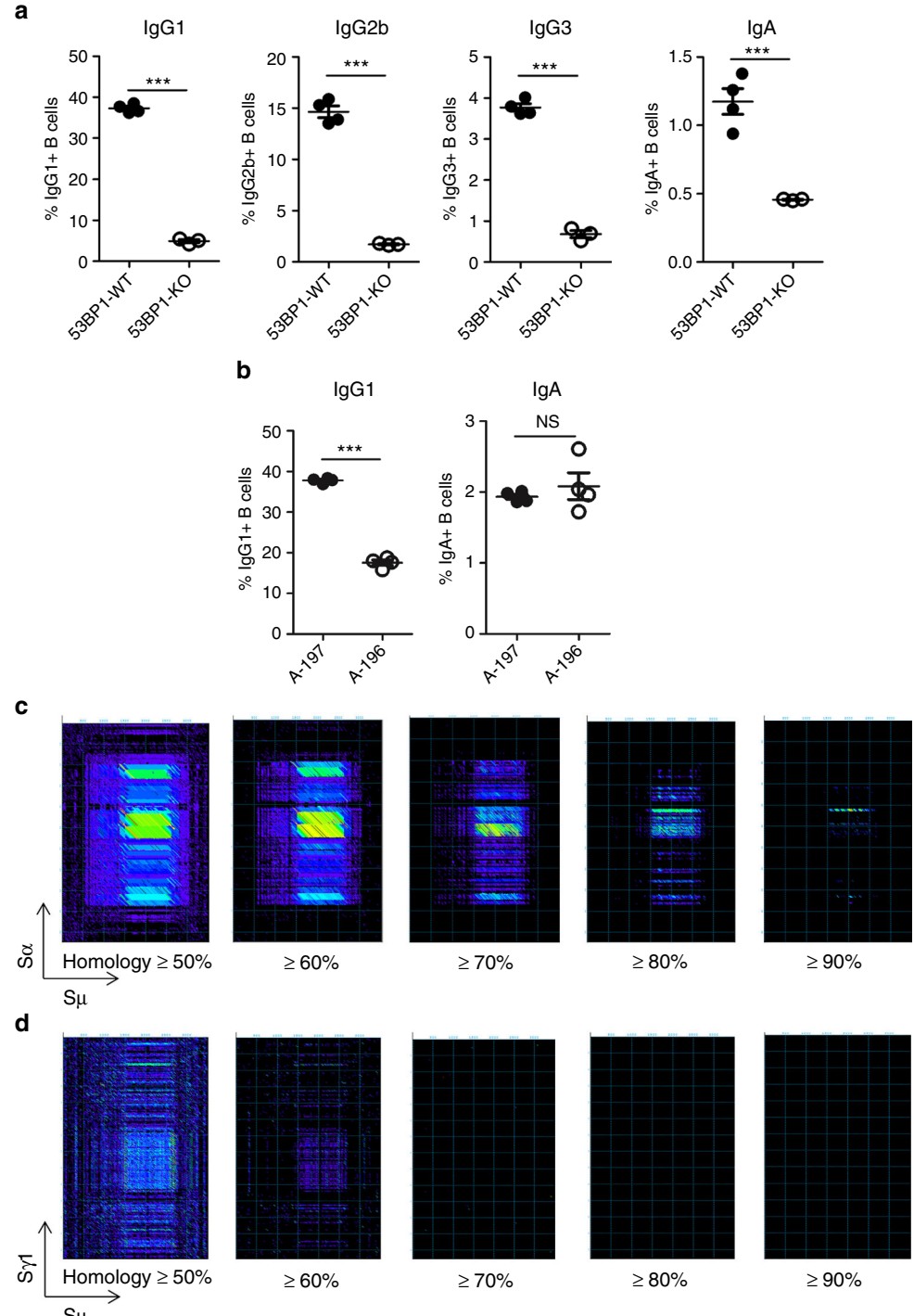

**Fig. 5** IgG1 CSR primarily relies on c-NHEJ, whereas IgA CSR is more dependent on A-EJ. **a** Spleen B cells isolated from 53BP1 KO or WT littermates were induced for ex vivo IgG1, IgG2b, IgG3, or IgA CSR for 4 days. 53BP1$^{-/-}$ B cells (deficient in c-NHEJ) exhibited a less severe defect in IgA CSR (~ 60% reduction) compared with IgG CSR (~ 80–90% reduction). Data represent two independent experiments each with three to four mice per group. **b** A-196 (10 μM) was used to inhibit c-NHEJ during the ex vivo CSR of WT spleen B cells and A-197 is the control compound. Inhibition of c-NHEJ markedly reduced IgG1 CSR, but not IgA CSR ($n = 4$ samples per group). Data represent three independent experiments. **c**, **d** The switch region sequence of Sμ on the IgH locus of C57BL/6 background mice were compared with that of **c** Sα or **d** Sγ1. Dot Matrix analysis of the mouse switch regions. The dots represent homologies between two switch regions with a search length of 30 bp and defined percentage of identity. Data in **a** and **b** were presented as mean ± SEM and were analyzed using two-tailed unpaired Student's $t$-test. ***$p < 0.001$; NS, not significant

various Ig isotypes except IgA in vivo. The ex vivo CSR assays revealed that Usp22 KO splenic B cells are defective in almost all the isotypes, except IgA. However, Usp22 KO LN B cells are defective in both IgG1 and, to a lesser extent, IgA CSR, which is similar to the CSR phenotype with 53BP1[−/−] mice. The discrepancy between these two studies is unclear. It is notable that CH12 B-cells have a B1 B-cell phenotype, whereas the vast majority of splenic B cells have a B2 B-cell phenotype[49,50], and recent studies have reported major differences between B1 and B2-B cells regarding their CSR toward IgA[50–52]. Nevertheless, as Usp22 deletion in vivo caused strong defects in various Ig isotypes but less so to IgA, Usp22 could be exploited as a therapeutic target in IgG or IgE-mediated autoimmune diseases, such as systemic lupus erythematosus, autoimmune thrombocytopenia, and asthma. It is expected that Usp22 inhibitors have the potential of ameliorating the pathogenesis of these antibody-mediated autoimmune diseases, without disturbing the homeostasis of the gut microbiota. Furthermore, our findings, that IgA CSR can utilize two DNA repair pathways while IgG CSR only employs one, highlight the biological need for IgA production as observed in μMT mice, which can produce IgA but not other isotypes[53].

In summary, we report that Usp22 is the first deubiquitinase that regulates both V(D)J recombination and CSR in vivo by promoting c-NHEJ. We found that IgG CSR is primarily mediated by c-NHEJ, while IgA CSR is more dependent on A-EJ, indicating that CSR to different isotypes involve distinct DNA repair pathways.

## Methods

**Mice.** C57BL/6 WT mice (Charles River Laboratory, St. Constant, QC, Canada; 6–8-week-old females) were bred in our Animal Vivarium facility at the University of Toronto. Mice containing two FRT sites in the upstream of exon 2 of Usp22 and two loxP sites flanking exon 2 of Usp22 were purchased from UC DAVIS KOMP Repository (Davis, CA) and bred with FLP-o deleter mice (The Jackson Laboratory), to generated Usp22[flox/+] mice. Usp22[flox/+] mice were then bred with CD19-Cre (The Jackson Laboratory) or Mb1-Cre mice, respectively. CD19-cre Usp22[flox/flox] mice were bred with Usp22[flox/flox] mice, to generate Usp22 KO (CD19-cre Usp22[flox/flox]) and WT littermates (i.e., Usp22[flox/flox]) for experimental use (sex-matched; 12–16 weeks old). Mb1-cre Usp22[flox/+] mice were bred with Usp22[flox/flox] mice, to generate Mb1-cre Usp22[flox/flox] (KO), Mb1-cre Usp22[flox/+] (het), and WT littermates (i.e., Usp22[flox/flox] or Usp22[flox/+]) for experimental use (sex-matched; 6–7 weeks old). 53BP1[+/−] mice were bred with 53BP1[+/−], to generate 53BP1[−/−] and WT littermates for experimental use (sex-matched; ~ 8 weeks old). All the mice were maintained under pathogen-free conditions. The experimental procedures were approved by the Animal Care Committee of University of Toronto.

**Usp22 and AID qPCR.** Spleen B cells were purified from CD19-cre Usp22 KO or WT littermates (around 12–16 weeks old), using the EasySep Mouse B-cell Isolation Kit (Stemcell Technologies). For LPS-stimulated group, spleen B cells (0.2 × 10[6]/mL) were cultured in the complete RPMI with 50 μg/mL LPS (Sigma Aldrich) for 3 days. RNA from resting or LPS-stimulated spleen B cells was extracted using TRIzol (Invitrogen) and then treated by DNaseI. The complementary DNA was prepared with the use of a SuperScript III reverse transcription kit (Life Technologies) and subjected to qPCR reactions of Usp22 or AID, with hypoxanthine guanine phosphoribosyl transferase (HPRT) as internal control. The sequences of the qPCR primers are described in the Supplementary Table 1.

**AID and H2Bub western blotting.** Purified spleen B cells were stimulated with 50 μg/mL LPS for 3 days and then collected for western blotting with mouse anti-AID monoclonal antibody (Cell Signaling Technology, clone: L7E7, catalog number: 4975; 1/1000 dilution) or rabbit polyclonal anti-β actin antibody (Sigma, catalog number: A2066; 1/1000 dilution). For H2Bub western blotting, the purified spleen B cells were first stimulated with LPS for 2.5 days, exposed to 8 Grays of γ-radiation, and then collected at various time points for western blotting with mouse monoclonal antibody against H2BK120Ub (Millipore, clone: 56, catalog number: 05-1312; 1/1000 dilution), or rabbit polyclonal antibody against total H2B (Abcam, catalog number: ab18977; 1/1000 dilution).

**Spleen and BM B-cell profiling.** Single-cell suspension of spleen was prepared from CD19-cre-Usp22 KO or WT littermate mice (around 12–16 weeks old), whereas BM cells were extracted from hind legs of Mb1-cre-Usp22 KO or WT

littermates (around 6–7 weeks old), followed by red blood cell lysis. The cells were strained and resuspended in fluorescence-activated cell sorting (FACS) buffer (2% fetal bovine serum in phosphate-buffered saline (PBS)), and incubated with mouse Fc blocker (2.4G2 mAb; 1/100 dilution). As previously described[54], splenic cells were stained with rat allophycocyanin (APC)-conjugated anti-CD45R/B220 (RA3–6B2; SouthernBiotech; 1/150 dilution), rat eFluor 450-conjugated IgM (II/41; eBioscience; 1/150 dilution), rat phycoerythrin (PE)-conjugated CD93 (AA4.1; eBioscience; 1/50 dilution), and rat fluorescein isothiocyanate (FITC)-conjugated CD23 (B3B4; eBioscience; 1/100 dilution); BM cells were stained with APC-conjugated anti-CD45R/B220, eFluor 450-conjugated IgM, rat PE Cy7-conjugated CD43 (S7; BD Pharmingen; 1/150 dilution), rat FITC-conjugated CD24 (30-F1; eBioscience; 1/100 dilution), and rat PE-conjugated BP-1 (6C3; BioLegend; 1/33 dilution). The stained cells were incubated with PI, acquired on a BD LSRFortessa X-20 flow cytometer (BD Biosciences), and analyzed by FlowJo software (Treestar Inc.).

**Total Ig ELISA assay.** Sera or fecal pellets were collected from CD19-cre Usp22 KO or WT littermate mice (around 7–8 weeks old), and fecal supernatant was prepared (10% wt/vol) with PBS. The Ig isotype-specific ELISA assays were performed as previously described[55]. Briefly, 96-well ELISA plates (NUNC) were coated with goat antibodies against mouse IgM (catalog number: 1021–01; 1/1000 dilution), IgG1 (catalog number: 1071–01; 1/1000 dilution), IgG2b (catalog number: 1090-01; 1/1000 dilution), IgG3 (catalog number: 1101-01; 1/333 dilution), or IgA (catalog number: 1040-01; 1/500 dilution) (all from SouthernBiotech), respectively, overnight at 4 °C. After blocking with 3% bovine serum albumin (BSA), 1/50-diluted sera or 1/5-diluted fecal samples were added into the wells, followed by serial fourfold dilution. The bound antibodies from sera or fecal supernatant were detected by the appropriate isotype-specific goat anti-mouse Ig that was conjugated with alkaline phosphatase (AP), IgM-AP (catalog number: 1021-04; 1/2000 dilution); IgG1-AP (catalog number: 1070-04; 1/2000 dilution); IgG2b-AP (catalog number: 1090-04; 1/2000 dilution); IgG3-AP (catalog number: 1100-04; 1/4000 dilution); IgA-AP (catalog number: 1040-04; 1/2000 dilution) (all from SouthernBiotech), followed by the development with SIGMAFAST p-Nitrophenyl phosphate substrate.

**Ex vivo CSR.** Spleen B cells were purified from CD19-cre Usp22 KO (around 12–16 weeks old), Mb1-cre Usp22 KO (around 6–7 weeks old), 53BP1[−/−] (around 6–10 weeks old) or their age-matched littermate control mice, using the EasySep Mouse B-cell Isolation Kit. To purify LN B cells, single-cell suspensions of peripheral LNs (inguinal, cervical, and axillary LN) and mesenteric LNs were stained with mouse BV605-conjugated anti-B220 (RA3-6B2; 1/200 dilution), and the sorting was performed with FACS Aria IIu cell sorter (BD Biosciences). Splenic or LN B cells (0.2 × 10[6]/mL) were cultured in complete RPMI with 50 μg/mL LPS for 4 days. For IgG1/IgE switching, 25 ng/mL of interleukin (IL)-4 (R&D Systems) was added; for IgG2b switching, 30 μg/mL dextran-sulfate (Sigma Aldrich) was added; for IgG3 switching, 3 or 10 ng/mL anti-IgD dextran (depending on the lots; Fina BioSolutions LLC) was added; for IgA switching, IL-4 (10 ng/mL), transforming growth factor (TGF)-β (2 ng/mL; R&D Systems), IL-5 (1.5 ng/mL; BD Pharmingen), and anti-IgD dextran (3 or 10 ng/mL; depending on the lots) were added. We noted that the variations in IgG3 or IgA CSR between different batches of experiments are mainly due to the various activities of different lots of anti-IgD dextran. For FACS analysis, cells were surface stained with PE rat anti-mouse IgG1 (A85-1, BD Pharmingen; 1/133 dilution), PE goat anti-mouse IgG2b (SouthernBiotech, catalog number: 1090-09; 1/133 dilution), PE goat anti-mouse IgA (SouthernBiotech, catalog number: 1040-09; 1/133 dilution), and FITC rat anti-mouse IgG3 (R40-82, BD Pharmingen; 1/100 dilution), whereas FITC rat anti-mouse IgE (R35-72, BD Pharmingen; 1/100 dilution) was intracellularly stained[56]. The stained cells were acquired on a FACSCalibur flow cytometer (BD Biosciences) and analyzed by FlowJo software.

For A-196 inhibition assay, splenic B cells were isolated from WT mice (around 6–10 weeks old) and cultured in complete RPMI media with the inhibitor, as previously described[35]. Spleen B cells (0.2 × 10[6]/mL) were first stimulated with LPS (25 μg/mL) in the presence of 10 μM A-196 or control compound A-197 for 1 day. For IgG1 CSR, the cells were changed with new media containing LPS (50 μg/mL) plus IL-4, and fresh A-196 or A-197 for 4 days. For IgA CSR, splenic B cells were stimulated with LPS (50 μg/mL), IL-4, TGF-β, IL-5, and anti-IgD dextran for 4 days.

**NP-specific ELISA and ELISPOT assay.** CD19-cre Usp22 KO or WT littermates (around 12–16 weeks old) were intraperitoneally immunized with 100 μg NP-CGG (Biosearch Technologies) in alum adjuvant (Thermo Scientific), as we previously described[22]. At day 22 post immunization, mice were bled for sera and killed for spleen collection. Sera from non-immunized or NP-immunized mice were added to ELISA plates that were coated with NP4-BSA (Biosearch Technologies; binds to high-affinity anti-NP antibodies) or NP32-BSA (Biosearch Technologies; binds to both high-affinity and low-affinity anti-NP antibodies), respectively, as we previously described[22]. Single-cell suspension of the spleen was prepared and subjected to NP-specific enzyme-linked immunosorbent assay spot (ELISPOT), in

which MultiScreen-HTS-HA filter plates were coated with NP32-BSA. Later steps were performed as previously described[22].

**Rotavirus-specific ELISA and ELISPOT assay.** CD19-cre Usp22 KO or WT littermates (around 12–16 weeks old) were infected with rotavirus as we previously described[23]. Fecal pellets were collected from each mouse one day before rotavirus challenge and on the indicated days post infection. Fecal supernatant was prepared (10% wt/vol) with PBS and ½ dilution was used for fecal anti-rotavirus IgA and rotavirus antigen ELISA assays, as previously described[23]. At day 23 post rotavirus infection, rotavirus-specific IgA+ ELISPOT assay was performed to measure the numbers of IgA+ RV-specific antibody-secreting cells from the BM and the lamina propria from the small intestine. Lymphocytes from the lamina propria were prepared by Percol gradient as we previously described[23]. BM cells were flushed out from the femurs and tibia of CD19-cre Usp22 KO or WT littermates, followed by red blood cell lysis. MultiScreen-HTS-HA filter plates were coated overnight with inactive rotavirus antigen (Microbix) and then blocked with complete RPMI media. The plates were incubated with twofold serial dilutions of BM cells or lymphocytes from lamina propria overnight, followed by detection with horseradish peroxidase (HRP)-conjugated goat anti-mouse-IgA Ab (SouthernBiotech, catalog number: 1040–05; 1/500 dilution). Then the colorimetric precipitating substrate for HRP, AEC (Vector Laboratories), was used to develop the plate. Positive dots on the membrane within wells were counted with a Nikon stereomicroscope.

**V(D)J and D-J PCR.** BM cells from Mb1-cre-Usp22 KO or control WT mice (around 6–7 weeks old) were stained with APC-conjugated anti-CD45R/B220, PE Cy7-conjugated CD43, FITC-conjugated CD24, and PE-conjugated BP-1 and PI. The sorting of Hardy Fraction B or C Pro-B cells were performed on a BD Influx Cell Sorter (BD Biosciences), followed by genomic DNA extraction using the proteinase K method. For the distal and proximal V(D)J PCR, genomic DNA from Fraction C of BM B cells was used and two rounds of PCR amplification were performed as previously described[28]: the first round of PCR reaction, containing both $V_H$J588 and $V_H$7183, and the $J_H$4E primer, was performed over 30 cycles (1 min at 95 °C, 1 min at 60 °C, and 1.5 min at 72 °C); for the second round of PCR, 1 µl aliquots of the first round PCR reaction was used, containing either $V_H$J588 or $V_H$7183, with the nested $J_H$4A primer (30 cycles: 1 min at 95 °C, 1 min at 63 °C, and 1.5 min at 72 °C). Glyceraldehyde 3-phosphate dehydrogenase (GAPDH) was used as loading control, with 25 PCR cycles (30 s at 94 °C, 30 s at 50 °C, and 2.5 min at 72 °C)[57]. The sequences of primers are described in the Supplementary Table 1.

**Cell cycle analysis and CFSE dilution assay.** The cell cycle analysis was performed with the use of Click-iT Plus EdU Alexa Fluor 647 kit (ThermoFisher Scientific). EdU is the modified analog of thymidine and gets incorporated into newly synthesized DNA. In brief, purified spleen B cells from CD19-cre Usp22 KO or WT littermates were stimulated with LPS for 3 days, before pulsing with Edu for 4 h. The cells were fixed with 4% formaldehyde, permeabilized with Perm buffer, and stained with Click-it reaction cocktail. After washing, the cells were treated with 100 µg/mL RNAse A for 10 min and then stained with PI. The stained cells were acquired on a BD LSRFortessa X-20 flow cytometer and analyzed by FlowJo software.

For CFSE dilution assay, purified spleen B cells were pulsed with 5 µM CFSE (Celltrace, Life Technologies), washed, and cultured in the presence of LPS plus IL-4. Cells were collected 4 days after CFSE pulsing and IgG1 CSR stimulation, stained with PE anti-mouse IgG1, and acquired with a BD LSRFortessa X-20 flow cytometer and analyzed by FlowJo software. The concentration of pulsed CFSE decreases as the cells divide, and thus the intensity of CFSE reflects the stages of cell division.

**γH2AX immunofluorescence staining.** Spleen B cells were purified from CD19-cre Usp22 KO or WT littermates, and stimulated with LPS for 2.5 days, before 0 or 8 Gy γ-irradiation (Gammacell 1000). At 1 h or 4 h post irradiation, the LPS-stimulated cells were cytospin onto poly-L-lysine slides, followed by Triton X-100 permealization. The slides were blocked with 10% BSA (BioShop) plus goat F(ab) polyclonal antibody against mouse IgG (Abcam, catalog number: ab6668; 1/100 dilution), to decrease nonspecific staining. The γH2AX staining using mouse monoclonal antibody against γH2AX (Millipore, clone: JBW301, catalog number: 05–636; 1/250 dilution) and quantification of foci formation were performed as we previously described[14].

**Germline transcripts and switch junction analysis.** For germline transcript PCR, splenic B cells from CD19-cre Usp22 KO or WT mice were induced for IgG1 or IgA CSR ex vivo for 2 days. RNA was extracted and cDNA was synthesized as described above. Semi-quantitative PCR was employed to amplify Iµ-Cµ, Iγ1-Cγ1, and Iα-Cα germline transcripts using the primers as previously described[4], and GAPDH was used as an internal control[14].

For switch junction analysis, spleen B cells from CD19-cre Usp22 KO or WT littermates were induced for IgG1 or IgA CSR ex vivo for 4 days. Genomic DNA from these cells was extracted by chloroform/isopropanol and precipitated by ethanol. Switch µ-α and µ-γ1 junctions were PCR amplified with Q5 polymerase (New England Biolabs) using primers as we and others described[14,58], then

subsequently cloned using Zero Blunt TOPO (Life Technologies). Sequencing was carried out at The Centre for Applied Genomics (Toronto, Canada), and switch junction sequences were analyzed as we previously described[14]. To quantify AID activity at switch regions, we repeated the same procedure for switch junction analysis but instead using primers targeting the region upstream of switch µ or γ1.

**Switch region homology analysis.** Mouse immunoglobulin gene sequences for C57BL/6 were obtained from NCBI database (NC_000078.6 Ch12 and GRCm38.p4 C57BL/6 J) and the switch regions, based on their highly repetitive nature were defined by dotplot analysis using the DNAStar software (percentage: 60 or 70, Window: 30 bp, Min Quality: 1). Switch regions for BALB/c mice were described previously[59]. The sequence homology of µ and downstream switch regions of C57BL/6 and BALB/c background mice were subsequently compared by the dot-plot analysis (percentage: 50–90, Window: 30 bp, Min Quality: 1).

**Statistical analysis.** Data were presented as mean ± SEM and were analyzed using two-tailed unpaired Student's t-test or two-way analysis of variance as indicated. *$p$ < 0.05, **$p$ < 0.01, and ***$p$ < 0.001.

**Data availability.** The data that support the findings of this study are available from the corresponding author upon request.

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

## Acknowledgements

We thank the Martin lab members for their advice and discussion during this study, Dr Stephen Li for the help in B-cell profiling analysis by flow cytometry, Dr Michael Reth for the Mb1-Cre mice, to Dr Daniel Durocher for the *53BP1*$^{-/-}$ mice. C.L. is a recipient of the Canadian Institutes of Health Research Postdoctoral Fellowship. This research is supported by a grant from the Canadian Institutes of Health Research (Grant number PJT153307) to A.M.

## Author Contributions

C.L. designed the experiments, performed the research, analyzed data, and wrote the manuscript. T.I., C.S., M.B., E.L., and A.L. performed the research. Q.P., L.D., and J.L.G. analyzed data and edited the manuscript. A.M designed the experiments, analyzed data, and wrote the manuscript.

## Additional information

**Competing interests:** The authors declare no competing interests.

