## [Peer Review File · Nature Communications]

Reviewers' comments:

Reviewer #1 (Remarks to the Author):

In this manuscript, Martin and colleagues characterize the role of the histone H2B deubiquitinase Usp22 in B cell development and class switch recombination. The authors generate floxed alleles of Usp22, and demonstrate that B cell specific deletion using Mb1-cre leads to a defect in pro-to pre-B cell development and recombination of distal V to DH segments. Deletion of Usp22 in more mature B cells (using CD19-cre) lead to a defect in CSR to all isotypes except IgA. They further demonstrate that IgG CSR is primarily dependent on canonical NHEJ while IgA switching relies more on microhomology-dependent end-joining, and that Usp22 loss leads to defects in gH2AX foci formation and NHEJ. The authors propose that CSR to different isotypes is reliant on defined end-joining pathways, and that Usp22 participates in the NHEJ phase of CSR.

The DNA repair pathways and factors that facilitate end-joining during CSR are yet to be fully elucidated. Thus, generating a genetic model to demonstrate that Usp22 influences both CSR and VDJ recombination is interesting. However, the authors have previously published in Cell Reports that Usp22 impairs CSR to IgA in CH12 cells. True, CH12 is a B cell line, but why is there such a discrepancy between the CH12 data and the primary B cell data? There is concern (see below) that the IgA CSR conditions on splenic B cells employed by the authors are not optimal. Additionally, even for the data that is presented here, the authors should address several issues outlined below.

1. In Fig. 1, the authors show that there is no significant alteration in the number of immature/mature and FO B cells in the spleen between wt and Usp22 k/o B cells. However, there is a trend towards reduced numbers. Since it is the FO B cells that primarily undergo CSR in culture, and B cell maturation could affect CSR in ex vivo assays, the authors should purify B cells from lymph nodes where most of the cells should be mature.

2. The observation that IgA CSR is not altered while CSR to other isotypes are is surprising. This is even more surprising given the authors earlier Cell Reports paper that Usp22 knock-down in CH12 cells leads to a defect in CSR to IgA. In the ex vivo assay shown in Fig. 2, are the authors even detecting IgA by FACS? Is there an AID k/o control to show that what the authors are quantifying as IgA+ cells are not some staining artifacts?

3. Figure 3 shows defective pro-to pre-B cell transition and a defect in distal V to DJ joining. This study is so poorly developed that it would be better off to just take this part out. Why is there a pro-to pre-B transition defect? Is there more cell death? Can a H-chain transgene rescue the defect? Why is there an alteration in distal vs proximal V gene usage? Is cell-cycle progression altered leading to an aberrant time-window during which the Igh loci undergoes VDJ recombination?

4. Fig. 4a, how do the authors know that the immunoreactive band is AID? Is there an AID k/o control?

5. Fig. 4-the authors show that there is no effect on Sg1 mutations. However, as Stavnezer and colleagues have shown earlier, Sg1 mutations are observed at a much lower frequency than Smu mutations. Are there Smu mutations in Usp22-deficient splenic B cells activated ex vivo under conditions that induce IgA CSR?

6. The authors should absolutely score for gH2AX foci in cells stimulated under conditions of IgA CSR.

7. Fig. 6. The authors show that Usp22 k/o does not affect Sm-Sg joins. But then the authors propose that Usp22 promotes c-NHEJ. If this were true, then there should be more MH-joints just

like what has been observed for Lig 4 k/o B cells. How do the authors explain this data?

8. The IgA CSR in Fig. 6A is very low and it is tough to interpret the results without either better stimulation conditions or at least an AID k/o control as background.

Reviewer #2 (Remarks to the Author):

Authors demonstrate the role the H2B deubiquitinase Usp22 on VDJ recombination and CSR to IgG and IgE. Usp22, in contrast, is not implicated in IgA CSR suggestion the use of the A-EJ pathway instead of the c-NHEJ pathway for DNA repair. These data are interesting for both the VDJ and CSR processes. Some experiments (but not all) are convincing. Numerous points should be addressed and/or discussed.

Major comments

1. Overall, the manuscript is lacking in how well the experiments are described. The number of experiments, the number of mice and the degree of significance (meaning of *) is never indicated. These information should be indicated for all figures.

2. Authors should take into account in their manuscript that CH12 B-cells have a B1 B-cell phenotype while the vast majority of splenic B cells have a B2 B-cell phenotype. The vast majority of IgA+ B cells of the digestive tract have also a B1 B-cell phenotype. In their supplementary Figure 1, authors show elevated levels (50% increase) of MZ B-cells in spleen of Usp22-KO mice; MZ B-cell having a B1b B-cell phenotype. Recently studies have reported major differences between B1 and B2-B cells concerning their CSR toward IgA (Kim et al. J Immunol 2016; Issaoui et al., Cell Mol Immunol 2017).

3. The percentage of AID mutation in Table 1 is unconvincing. Why investigating mutation frequency upstream Sy1? Mutation levels are too low compared to other studies and surely not different from those found in AID-deficient mice. An higher mutation frequency in LPS stimulated B cells compared to resting B-cells would be of interest to strengthen the conclusion but mutation frequency in S μ -Sy1 or an AID-ChIP would be better.

4. The authors state that there is no difference in mRNA levels in Figure 4a. There is a difference, but it is not significant. What is the precise significance? How many mice per experiments? Two or three new mice per group might change the significance.

5. The hypothesis between the c-NHEJ vs A-EJ pathways is based apparently on 4 junctions with high microhomology length. The data is hard to interpret or even understand based on the limited information provided. How many mice for these experiments? The location of breakpoints in switch regions would be of interest especially is these junctions located around the same place (bias related to a single mouse?).

Minor comments

1. In the introduction some sentences explaining differences between c-NHEJ and A-EJ would be of interest for the average reader.

2. At the end of their introduction authors suggest that "CSR to different isotype may involve distinct DNA repair pathways". The preferential use of the A-EJ pathways has been already reported for CSR toward IgD (Ghazzoui et al., Immunol Lett 2017; Issaoui et al., Cell Mol Immunol 2017).

3. Figure 1C: Is there a difference between Usp22 mRNA levels before and after LPS stimulation?
4. Figure 2d and 2e: How author explain such decrease of IgG3 and IgA CSR between CD19-cre and Mb1-cre mice?
5. Sentences explaining differences between Fraction A, B and C would be of interest.
6. Figure 3C (left part): Why the 1/2 PCR dilution for Usp22 KO mice is higher than the undiluted PCR product?
7. For Usp22 KO mice there is more cells in S phase at 2.5 days (Figure 4C, right part) but no cell increase at day 3 (Figure 4d, left part). How to explain these results?
8. Figure 4e: More explanation would be of interest.
9. An hypothesis to explain elevated AID in spleen of Usp22 KO mice?
10. Figure 4b. How to explain the decrease at 1 hour?
11. Please, explain Edu for the readers.
12. Figure 5: Is there a difference for the type of insertion?
13. Figures 2 and 6: 1% of IgA CSR for 53BP1-WT (2% with A197), 2% for Mb1-cre Usp22-WT and 5% for CD19-cre Usp22-WT. How to explain such differences? Is a decrease from 1% to 0.5% really physiologically significant?

We thank our Reviewers and Editor for their suggestions and encouraging comments. We have amended our manuscript in response to all these suggestions/comments. The responses and changes (red font in the manuscript) are detailed below:

Reviewer #1 (Remarks to the Author):

The DNA repair pathways and factors that facilitate end-joining during CSR are yet to be fully elucidated. Thus, generating a genetic model to demonstrate that Usp22 influences both CSR and VDJ recombination is interesting. However, the authors have previously published in *Cell Reports* that Usp22 impairs CSR to IgA in CH12 cells. True, CH12 is a B cell line, but why is there such a discrepancy between the CH12 data and the primary B cell data? There is concern (see below) that the IgA CSR conditions on splenic B cells employed by the authors are not optimal.

Response: As the Reviewer pointed out, our previous *Cell Reports* paper showed that there is a defect in IgA CSR in CH12 when Usp22 is knocked down or out. In this current manuscript, we found that most of the CH12 phenotype holds up in the Usp22 KO mice, as Usp22 KO splenic B cells are defective in CSR to all the isotypes except IgA. Following the Reviewer's suggestion below (**point 1**), we also performed the *ex vivo* CSR with Usp22-WT or -KO lymph nodes (LN) B cells, and found that Usp22 KO LN B cells are defective in both IgG1 and IgA CSR, although Usp22 LN B cells exhibited a more severe defect in IgG1 CSR than that in IgA CSR when compared Usp22 littermate control (please see **point 1** for details). Thus, the difference between CH12 and murine primary B cells is not dramatic. Furthermore, as Reviewer 2 pointed out, CH12 B-cells have a B1 B-cell phenotype while the vast majority of splenic B cells have a B2 B-cell phenotype, and MZ B-cell having a B1b B-cell phenotype. Recent studies have reported major differences between B1 and B2-B cells concerning their CSR toward IgA (Kim et al. *J Immunol* 2016; Issaoui et al., *Cell Mol Immunol* 2017). We have included this information in the revised manuscript.

Regarding the IgA CSR conditions, we found that the variations in IgA CSR between different batches of experiments are mainly due to the various activities of anti-IgD dextran of different lots (we added this information in the revised Methods and Materials). But using the AID k/o as control, we validated that our FACS detection of IgA CSR is specific (please see **point 2** for details).

1. In Fig. 1, the authors show that there is no significant alteration in the number of immature/mature and FO B cells in the spleen between wt and Usp22 k/o B cells. However, there is a trend towards reduced numbers. Since it is the FO B cells that primarily undergo CSR in culture, and B cell maturation could affect CSR in *ex vivo* assays, the authors should purify B cells from lymph nodes where most of the cells should be mature.

Response: We thank the Reviewer for the suggestion. As discussed above, we have followed this advice and sorted for B220⁺ B cells from peripheral lymph nodes (inguinal, cervical and axillary LN) and mesenteric lymph nodes, and then perform the *ex vivo* IgG1 and IgA CSR, respectively. We found that Usp22 LN B cells exhibited a more severe defect in IgG1 CSR than that in IgA CSR when compared Usp22

littermate control, which is similar to our CSR phenotype with 53BP1^{-/-} mice, further suggesting that Usp22 is c-NHEJ factor *in vivo* and CSR to different isotypes involves distinct DNA repair pathways. We have included these data in the revised manuscript.

2. The observation that IgA CSR is not altered while CSR to other isotypes are is surprising. This is even more surprising given the authors earlier Cell Reports paper that Usp22 knock-down in CH12 cells leads to a defect in CSR to IgA. In the ex vivo assay shown in Fig. 2, are the authors even detecting IgA by FACS? Is there an AID k/o control to show that what the authors are quantifying as IgA⁺ cells are not some staining artifacts?

Response: We do acknowledge that our IgA CSR is lower than other papers, which is probably due to different sources, lots and/or concentrations of stimulants. We have tested the different lots/concentrations of anti-IgD dextran, and found that IgA CSR is highly variable depending on the lot of anti-IgD dextran (most of the IgA ex vivo CSR data in this manuscript were performed with the 2nd batch of anti-IgD dextran). Nonetheless, by using the AID k/o as control, we validated that our FACS detection of IgA CSR ex vivo is specific, under each batch of anti-IgD dextran stimulation (**Figure R1**). In the revised manuscript, we have included the representative IgA FACS plots with AID k/o controls.

Figure R1: Splenic B cell in vitro CSR using different batches of anti-IgD dextran shows different effects on IgA CSR.

3. Figure 3 shows defective pro-to pre-B cell transition and a defect in distal V to DJ joining. This study is so poorly developed that it would be better off to just take this part out. Why is there a pro-to pre-B transition defect? Is there more cell death? Can a H-chain transgene rescue the defect? Why is there an alteration in distal vs proximal V gene usage? Is cell-cycle progression altered leading to an aberrant time-window during which the Igh loci undergoes VDJ recombination?

Response: We appreciate the Reviewer's concern. We think that it is important to keep the B cell development and distal V to DJ data in the manuscript to maintain the breadth of this study. The V(D)J

data in Mb1-cre-Usp22 KO mice are consistent with our previous V(D)J data with A70.2 cell line in **Cell Reports** suggested that SAGA complex, in which Usp22 is the deubiquitinase module, is important for V(D)J recombination.

However, we acknowledge that more work can be done to further validate these findings, and many of the suggested experiments raised by the reviewer would further strengthen the conclusions. However, many of these experiments (especially the H-chain transgene rescue experiment) would take considerable time. Instead of removing the data, as the reviewer suggested, we have moved all of this data to the supplementary section, and deleted the paragraph of B cell development in the revised manuscript. Instead, we mentioned briefly in the CSR section of Mb1-cre-Usp22 KO mice that these KO mice also exhibit defects in B cell development and possibly V(D)J recombination. We also included some of the Reviewer's points as future studies in the Discussion.

Nevertheless, if the Reviewer and Editor insisted that this part should be removed from the manuscript, we would be happy to do so.

4. Fig. 4a, how do the authors know that the immunoreactive band is AID? Is there an AID k/o control?

Response: In the revised manuscript, we now show another representative blot with AID k/o control in Figure 3a (formerly Fig. 4a), to confirm that the band is specific for AID. Furthermore, the anti-AID monoclonal antibody (Cell Signalling, clone: L7E7) that we used is well validated, and we have published a number of papers with it. We included some of this information in the revised Methods and Materials.

5. Fig. 4-the authors show that there is no effect on S μ mutations. However, as Stavnezer and colleagues have shown earlier, S μ mutations are observed at a much lower frequency than S μ mutations. Are there S μ mutations in Usp22-deficient splenic B cells activated ex vivo under conditions that induce IgA CSR?

Response: As the Reviewer suggested, we have performed this experiment and found that the frequency of S μ mutations is indeed much higher (around 3-4 fold) than that of S γ 1 mutations. However, consistent with our S γ 1 mutation data, we found a comparable frequency of S μ mutations between IgA-switched Usp22 WT and KO cells. We have included these data in the revised manuscript.

6. The authors should absolutely score for γ H2AX foci in cells stimulated under conditions of IgA CSR.

Response: Following the Reviewer's advice, we have scored the γ H2AX foci pre- and post-irradiation in WT vs Usp22-KO splenic B cells under IgA CSR conditions. In unirradiated cells, we find that 60 hours after we induced for IgG3 CSR (LPS) and IgA CSR (LPS + IL4 + IL5 + TGF β + anti-IgD dextran), there was increased γ H2AX formation in Usp22-KO cells compared to controls, suggesting that there is increased DSBs possibly due to reduced repair in the Usp22-KO cells. When cells are irradiated, we found a variable effect on γ H2AX formation in Usp22-KO cells compared to controls, but a striking reduction 4h post irradiation, a result that is similar to those obtained in our previous report using CH12F3-2 cells.

We have included the data in the revised manuscript.

7. Fig. 6. The authors show that Usp22 k/o does not affect Sm-Sg joins. But then the authors propose that Usp22 promotes c-NHEJ. If this were true, then there should be more MH-joints just like what has been observed for Lig 4 k/o B cells. How do the authors explain this data?

Response: Our data suggest that IgG1 CSR does not use alternative end joining (A-EJ) as efficiently as IgA CSR when c-NHEJ is impaired, which is supported by our observations in MH length of usage between Usp22 WT and KO and homology analysis of switch regions. Actually, we indeed observed a trend of increased 6+bp MH usage in Usp22-KO IgG1 group (9%) when compared to Usp22-WT IgG1 group (2%), although the average MH usage is comparable between these groups. We also can not exclude the possibility that although Usp22 promotes c-NHEJ, its role in c-NHEJ is not as critical as other c-NHEJ core factors, such as Lig 4 or 53BP1. In addition, it has been shown that different c-NHEJ defects seem to result in different phenotype for $\Sigma\mu$ - $\Sigma\gamma$ joins, in terms of increases in MH, increased insertion, or increased sequential switching. We have included some of this information in the revised manuscript.

8. The IgA CSR in Fig. 6A is very low and it is tough to interpret the results without either better stimulation conditions or at least an AID k/o control as background.

Response: As described in the above response to Reviewer One (**point 2**), IgA CSR is highly variable depending on the lot of anti-IgD dextran. Nonetheless, with AID k/o cells as control, we validated the specificity of our IgA detection.

Reviewer #2 (Remarks to the Author):

Authors demonstrate the role the H2B deubiquitinase Usp22 on VDJ recombination and CSR to IgG and IgE. Usp22, in contrast, is not implicated in IgA CSR suggestion the use of the A-EJ pathway instead of the c-NHEJ pathway for DNA repair. These data are interesting for both the VDJ and CSR processes. Some experiments (but not all) are convincing. Numerous points should be addressed and/or discussed.

Major comments

1. Overall, the manuscript is lacking in how well the experiments are described. The number of experiments, the number of mice and the degree of significance (meaning of *) is never indicated. These information should be indicated for all figures.

Response: We thank the Reviewer for this suggestion. We have included this information in the methods and figure legends.

2. Authors should take into account in their manuscript that CH12 B-cells have a B1 B-cell phenotype while the vast majority of splenic B cells have a B2 B-cell phenotype. The vast majority of IgA+ B cells of the digestive tract have also a B1 B-cell phenotype. In their supplementary Figure 1, authors show

elevated levels (50% increase) of MZ B-cells in spleen of Usp22-KO mice; MZ B-cell having a B1b B-cell phenotype. Recently studies have reported major differences between B1 and B2-B cells concerning their CSR toward IgA (Kim et al. J Immunol 2016; Issaoui et al., Cell Mol Immunol 2017).

Response: We thank the Reviewer for the suggestion. Indeed, as discussed above in response to Reviewer One's comments (**point 1**), we also performed the *ex vivo* CSR with Usp22-WT or -KO lymph nodes (LN) B cells, and found that Usp22 KO LN B cells are defective in both IgG1 and IgA CSR, although Usp22 LN B cells exhibited a more severe defect in IgG1 CSR than that in IgA CSR when compared Usp22 littermate control. Thus, the difference between CH12 and murine primary B cells is not dramatic. We have included this information in the revised manuscript.

3. The percentage of AID mutation in Table 1 is unconvincing. Why investigating mutation frequency upstream S γ 1? Mutation levels are too low compared to other studies and surely not different from those found in AID-deficient mice. An higher mutation frequency in LPS stimulated B cells compared to resting B-cells would be of interest to strengthen the conclusion but mutation frequency in S μ -S γ 1 or an AID-ChIP would be better.

Response: The reason for why we investigated the mutation frequency upstream S γ 1 is to exclude the possibility that the impaired IgG1 CSR in Usp22 k/o group is due to the reduced S γ 1 mutation. We agree with the Reviewer that we should also have investigated the S μ mutation frequency, as was also suggested by Reviewer One. Hence, we analyzed mutations at the S μ region under IgA CSR conditions, and found a comparable frequency of S μ mutations between IgA-switched Usp22 WT and KO cells. We have included these data in the revised manuscript.

4. The authors state that there is no difference in mRNA levels in Figure 4a. There is a difference, but it is not significant. What is the precise significance? How many mice per experiments? Two or three new mice per group might change the significance.

Response: It was 3 mice per group. We followed the Reviewer's suggestion, by adding another 2 new mice per group, and now the difference reached is statistically significant. We have updated this figure in the revised manuscript.

5. The hypothesis between the c-NHEJ vs A-EJ pathways is based apparently on 4 junctions with high microhomology length. The data is hard to interpret or even understand based on the limited information provided. How many mice for these experiments? The location of breakpoints in switch regions would be of interest especially if these junctions located around the same place (bias related to a single mouse?).

Response: We analyzed three mice per group for these experiments, and around 50-60 unique switch junction sequences per group were analyzed. This information has been added to the revised manuscript. We have also added, as suggested by the reviewer, the location of breakpoints in switch

regions in the revised manuscript. There is no obvious difference in the distribution of breakpoints across μ , α , or $\gamma 1$ switch regions between Usp22-WT and -KO group.

Minor comments

1. In the introduction some sentences explaining differences between c-NHEJ and A-EJ would be of interest for the average reader.

Response: We have now included this information in the revised Introduction.

2. At the end of their introduction authors suggest that “CSR to different isotype may involve distinct DNA repair pathways”. The preferential use of the A-EJ pathways has been already reported for CSR toward IgD (Ghazzaoui et al., Immunol Lett 2017; Issaoui et al., Cell Mol Immunol 2017).

Response: We thank the Reviewer for this information and have included these references in the revised manuscript.

3. Figure 1C: Is there a difference between Usp22 mRNA levels before and after LPS stimulation?

Response: Yes, there is a significant reduction in Usp22 mRNA level after LPS stimulation, and we have included this data in the revised manuscript.

4. Figure 2d and 2e: How author explain such decrease of IgG3 and IgA CSR between CD19-cre and Mb1-cre mice?

Response: These differences were mainly due to the different lots of anti-IgD dextran (see **Figure R1** above, and **Figure R2** below), as the CD19-cre and Mb1-cre mice were carried out at different times. However, we performed all CSR assays for WT and KO mice for each cre-expressing mice at the same time.

Figure R2: Splenic B cell in vitro CSR using different batches of anti-IgD dextran shows different effects on IgG3 CSR.

5. Sentences explaining differences between Fraction A, B and C would be of interest.

Response: We have now provided more information about these fractions in the revised manuscript.

6. Figure 3C (left part): Why the ½ PCR dilution for Usp22 KO mice is higher than the undiluted PCR product?

Response: It is likely due to DNA migration variation in that lane. Based on the comments of Reviewer One, we have moved this figure to the Supplementary section.

7. For Usp22 KO mice there is more cells in S phase at 2.5 days (Figure 4C, right part) but no cell increase at day 3 (Figure 4d, left part). How to explain these results?

Response: For the cell cycle analysis, the cells were stimulated with LPS for 3 days before an Edu pulse. There is a trend of increased cell proliferation in Usp22 KO group when compared to Usp22 WT control, although the difference is not statistically significant. Usp22 KO group exhibited a more dramatic increase in cell proliferation at d4 post-LPS stimulation.

8. Figure 4e: More explanation would be of interest.

Response: We have provided more information about the CFSE assay in the revised manuscript.

9. An hypothesis to explain elevated AID in spleen of Usp22 KO mice?

Response: It has been demonstrated that H2B ubiquitination is involved in regulating gene transcription. We hypothesize is that H2B ubiquitination enhance transcription of certain genes, such as *Aicda*. We have included discussion of this point in the revised manuscript.

10. Figure 4b. How to explain the decrease at 1 hour?

Response: Excessive H2B ubiquitination may decrease the recruitment/stability of H2AX kinase (i.e. ATM and DNAPK) possibly via sterical hindrance, or/and enhance the recruitment of certain γ H2AX phosphatase. We have discussed these possibilities in the revised manuscript.

11. Please, explain Edu for the readers.

Response: We have explained Edu in the revised manuscript.

12. Figure 5: Is there a difference for the type of insertion?

Response: When compared to WT control, there is a slight decrease in the percentage of insertion in Usp22 KO group (IgG1: from 28% to 21%; IgA: from 19% to 11%). However, these results were not statistically significant.

13. Figures 2 and 6: 1% of IgA CSR for 53BP1-WT (2% with A197), 2% for Mb1-cre Usp22-WT and 5% for CD19-cre Usp22-WT. How to explain such differences? Is a decrease from 1% to 0.5% really physiologically significant?

Response: The IgA CSR variation between experiments is mainly due to different lots of anti-IgD dextran, as we showed above (**Figure R1**). Using the AID KO cells as the control, we have validated the specificity of our IgA FACS detection. Importantly, we performed all the CSR of related groups (e.g. WT vs KO, A-197 vs A-196) at the same time.

REVIEWERS' COMMENTS:

Reviewer #1 (Remarks to the Author):

The authors have addressed most of the queries raised by this reviewer. Not all the issues have been experimentally addressed but there has been an honest attempt to address all the concerns. This is an interesting study and should be published.

Reviewer #2 (Remarks to the Author):

No new comments. The authors responded adequately to the critics.